# Development of Methamphetamine Conjugated Vaccine through Hapten Design: In Vitro and In Vivo Characterization

**DOI:** 10.3390/vaccines11020340

**Published:** 2023-02-02

**Authors:** Md Kamal Hossain, Majid Davidson, Jack Feehan, George Deraos, Kulmira Nurgali, John Matsoukas, Vasso Apostolopoulos

**Affiliations:** 1Institute for Health and Sport, Victoria University, Melbourne, VIC 3030, Australia; 2Regenerative Medicine and Stem Cells Program, Australian Institute for Musculoskeletal Science (AIMSS), Melbourne, VIC 3021, Australia; 3Immunology Program, Australian Institute for Musculoskeletal Science (AIMSS), Melbourne, VIC 3021, Australia; 4New Drug, Patras Science Park, 26500 Patras, Greece; 5Department of Medicine Western Health, University of Melbourne, Melbourne, VIC 3010, Australia; 6Department of Physiology and Pharmacology, Cumming School of Medicine, University of Calgary, Alberta, AB T2N 4N1, Canada

**Keywords:** methamphetamine, oxidized mannan, hapten, peptide linker, antibody, stability, vaccine, ice, drug addiction

## Abstract

Background: Methamphetamine (METH) substance-use disorder is an ever-growing global health issue with no effective treatment. Anti-METH vaccines are under investigation as an alternative to existing psychological interventions. This platform has made significant progress over past decades mainly in preclinical stages, and efforts to develop an anti-METH vaccine with a high antibody response are of utmost importance. Methodology: A novel conjugated anti-METH vaccine was developed using METH HCl as the starting material for the design of hapten, a peptide linker consisting of five lysines and five glycines, and finally immunogenic carrier mannan, which is novel to this platform. All the chemical reaction steps were confirmed by several analytical techniques, and the immunogenicity of the developed vaccine was investigated in a mouse model. Results: Thin-layer chromatography and gas chromatography confirmed the reaction between METH and peptide linker. UV, NMR and color tests were used to confirm the presence of the aldehyde groups in oxidized mannan (OM). The final conjugated vaccine was confirmed by UV and LC-MS. The stability of mannan, the METH hapten, and the final vaccine was evaluated by UV and LC-MS and demonstrated satisfactory stability over 3 months in various storage conditions. Animal studies supported the immunogenicity of the novel vaccine. Conclusions: We successfully developed and characterized a novel METH vaccine in vitro and in vivo. The present study findings are encouraging and will form the basis of further exploration to assess its effectiveness to prevent METH addiction in preclinical models.

## 1. Introduction

Methamphetamine (METH) substance-use disorder (SUD) is a complex neurological condition with no effective treatment or management options [1,2,3,4]. Psychological interventions such as cognitive behavioral therapy (CBT), matrix models, contingency management and residential rehabilitation are the main management options for most SUDs including METH [1,5,6,7]. However, these psychological interventions have limited effectiveness and face several challenges such as a high relapse rate after rehabilitation, limited access and significant variation in outcomes [3,8,9]. METH influences the release of a range of neurotransmitters such as dopamine, serotonin, adrenaline and norepinephrine, making pharmacological interventions focusing on receptor modulation impossible [10,11,12,13]. The limitations of the current management strategies have necessitated the development of more effective interventions that can be used alongside psychological interventions to achieve a lasting impact [14]. Immunotherapies against substances of abuse including METH have gained attention due to their potential for use as both prophylactic and therapeutic treatments. Immunotherapies such as anti-METH vaccines and monoclonal antibodies (mAbs) can bind to the drug substance in the circulation, blocking its action without affecting the receptor, and are thus considered safer compared to other pharmacological interventions [15]. These approaches, as an adjunct to standard psychotherapies, have great potential for the management and treatment of METH SUD [8,16].

METH is a small molecule with a molecular weight of 149 Daltons and so lacks an immunogenic effect on its own. Generally, only compounds with molecular weights greater than 10,000 Daltons are able to elicit an antibody response [10,17,18]. Therefore, to create an immunotherapy, METH must be conjugated to a large immunogenic carrier to be recognized by the immune system and stimulate antibody responses [14,19]. Several immunogenic protein carriers, such as keyhole limpet hemocyanin (KLH), bovine serum albumin (BSA), ovalbumin (OVA), tetanus toxoid (TT) and diphtheria toxoid (DT), have been investigated since the inception of anti-drug vaccines including against METH [10,20,21]. METH haptens are developed through a design strategy where an immunologically inactive linker is used to connect the drug of interest to its protein carrier. Usually, METH or amphetamine is used to design an appropriate hapten by introducing the linker at the various positions of either molecule [8]. Amphetamine is more commonly used to synthesize METH haptens as it is easy to modify the primary amine to introduce a linker [22,23,24]. However, people more commonly abuse METH due to its strong euphoric effect, making it more rational to use METH as the starting material during hapten synthesis to ensure better recognition/specificity of the immune response. In most earlier studies, emphasis was given to preserving the native structure of the compound during hapten design to ensure better specificity [8,14,15,19,24,25].

In earlier studies, hapten synthesis was performed through alkylation and short-length peptide linkers. However, numerous studies have shown that a combination of five lysines and five glycines (known as (KG)_5_) has the optimum molecular length to elicit antibody responses when used with oxidized mannan (OM). Mannan is a polysaccharide extracted from yeast, which is non-toxic, natural and biodegradable and has been investigated in vaccine development due to its immunogenic properties [26]. Mannan can produce an antibody response through activation of the TLR4 and mannose receptors in antigen-presenting cells (APCs) [27,28]. Mannan was used as an immunogenic carrier more than 25 years ago [29], and the steps in its development were characterized by various techniques such as sodium dodecyl sulfate–polyacrylamide gel electrophoresis (SDS-PAGE), capillary electrophoresis (CE), ultraviolet (UV) spectrometry and electrochemical voltammetry [30]. The METH–(KG)_5_–OM conjugate vaccine is a novel platform, so characterizing each of the steps in its development, including hapten design, conjugation with OM, the immunogenic response in animal models and stability of the vaccine components, is crucial.

We investigated the development of an anti-METH conjugated vaccine using METH HCl as the starting material, [(KG)_5_] as the peptide linker and oxidized mannan as the immunogenic carrier. Further, we evaluated the stability of the three major elements of the vaccine formulation: the hapten, OM and the final vaccine [hapten–(KG)_5_–OM]. Finally, the vaccine was evaluated in mice for its immunogenic potential.

## 2. Materials and Methods

### 2.1. Chemicals

METH hydrochloride (Ref D816) was procured from NMI, Australia. The identity of the purchased METH was confirmed by gas chromatography–mass spectrometry (GC-MS). Mannan, 2,4-dinitrophenylhydrazone, collidine, sodium periodate (NaIO_4_), 1-hydroxy-7-azabenzotriazole (HOAt), dithiothreitol (DTT), 2-(1H-benzotriazole-1-yl)-1,1,3,3-tetramethylaminium tetrafluoroborate (TBTU) and Ninhydrin^®^ were purchased from Sigma Aldrich, (Australia) and used without further purification. The peptide linker [(KG)_5_] with a purity of >85% was procured from Synpeptide (Shanghai, China). All other reagents and chemicals used in this study were of analytical standard and used without further potency adjustment or purification.

### 2.2. Synthesis of Hapten Using METH HCl and (KG)_5_ Linker

Briefly, the procedure of hapten synthesis is as follows. First, 2 mg of METH HCl and 5 mg of TBTU were mixed in 2 mL dimethylformamide (DMF). Then, 10 mg (KG)_5_, 2.04 mg HOAt and 3.64 mg collidine were dissolved in a 2 mL DMF solution. The (KG)_5_ solution was added to the METH solution drop-by-drop at room temperature (RT) before incubation at 25 ^o^C for 24 h. The reaction between METH and (KG)_5_ was confirmed by GC (GCMS-QP2010, Shimadzu, Kyoto, Japan) and thin-layer chromatography (TLC, silica gel 60G F254 plate, Sigma-Aldrich, St. Louis, MO, USA). At the end of the reaction, 10 mL of acetonitrile was added to the reaction mixture, and the solution was freeze-dried (CHRIST, Alpha 1–4 LSC plus, Osterode am Harz, Germany) for 20 h. After the freeze-drying, water was added to the powder to precipitate the target compound for 24 h at 4°C. The solution was centrifuged at 4000 rpm for 20 min, and supernatant was discarded, with the obtained precipitate freeze-dried again. The Boc groups were removed by using TFA and DCM (95%/5% *v*/*v*) to free the amino group for further reaction with aldehydes in OM. The above solution was stored for 5 h at RT for deprotection, and the final solution was freeze-dried. The freeze-dried powder was precipitated by using diethyl ether to obtain the title compound. The solution was centrifuged at 4000 rpm for 20 min, and the precipitate was freeze-dried to obtain the final product. The identity of the final hapten was confirmed by LC-MS (Shimadzu LC-MS-2010 EV, Shimadzu, Kyoto, Japan) based on the molecular weight. The schematic diagram of the overall reaction protocol is shown in Figure 1.

### 2.3. Assessment of Reaction between METH and (KG)_5_ by TLC and GC

TLC and GC techniques were applied to confirm the reaction between (KG)_5_ and METH. The absence of the METH peak from the reaction mixture was the indicator of reaction completion. The TLC mobile phase consisted of 25% ammonia and methanol (1:25 ratio). METH solution (5 µL, solution A) was plated (silica gel 60G F254 plate, Sigma-Aldrich, St. Louis, MO, USA) and placed in the mobile phase tank for 10 min. The plate was taken out from the TLC mobile phase tank and air-dried at RT for 5 min. After air-drying, the plate was sprayed with 1% Ninhydrin solution, air-dried for 20 min and finally oven-dried at 100 °C for 30 min for color development. Similarly, after 24 h of reaction, 5 µL of solutions A + B was run through TLC to confirm the absence of METH in the METH–(KG)_5_ reaction mixture. Similarly, 50 µL of solution A (METH) and 100 µL of mixed solution were analyzed by GC to confirm the disappearance of METH in the METH–(KG)_5_ reaction mixture. The sensitivity of both TLC and GC methods was verified to ensure that the methods are suitable for the intended purposes.

### 2.4. Confirmation of METH Hapten Identity by LC-MS

The identity of the METH hapten [METH–(KG)_5_] was confirmed by LC-MS (Shimadzu LC-MS-2010 EV, Shimadzu, Kyoto, Japan). The METH hapten sample was assessed in both selective ion mode (SIM) and scan mode. The optimized instrument (LC-MS) operating conditions are as follows. The mobile phase A consisted of 0.1% TFA in water, and mobile phase B consisted of 90% acetonitrile+ 10% of 0.1% TFA in water. The flow rate was 0.2 mL/min. The LC time program was 0.01 min mobile B 68%, at 13 min mobile phase B 83% and at 17 min mobile phase B 68%. The run time was 17 min. The MS specification for Boc deprotected peptide was 943.19(+) and 942.19(−) and for METH–(KG)5:1074.42(+) and 1073.42(−).

### 2.5. Preparation of Oxidized Mannan

Mannan does not have any functional groups which react with amines in the hapten. Therefore, mannan was oxidized by using a method reported previously [31]. Initially, 14 mg mannan was dissolved in 1 mL of sodium phosphate buffer (pH 6.0), then 100 µL of 0.1 M NaIO_4_ was added to the above solution, and the solution mixture was incubated in a dark room for 1 h at 4 °C. Next, 10 µL of ethylene glycol was mixed with the above solution and incubated for 30 min in a dark room at 4 °C. The solution was then passed through a Sephadex^TM^ (Sigma-Aldrich, St. Louis, MO, USA) G-25M column (PD-10 column) equilibrated in bicarbonate buffer. Initially, 1.110 mL of sample was passed through the column (discarded), then 1.5 mL bicarbonate buffer (pH 9.0) was passed through (discarded) followed by 2 mL bicarbonate passed through and collected as OM (2 mL).

#### 2.5.1. Confirmation of Aldehyde Groups in OM by 2,4-DNPH Test

Aldehydes react with 2,4-dinitrophenylhydrazine (2,4-DNPH) reagent and form yellow, orange or reddish-orange precipitates. 2,4-DNPH (1 g) was added to 8.75 mL water and 23.25 mL ethanol in a volumetric flask. The mixture was placed in an ice bath and stirred. Then, 5 mL of concentrated sulfuric acid was added slowly and stirred on a hot plate (60 °C) until all 2,4-DNPH was dissolved. Mannan and OM were added to a TLC silica plate and soaked in the 2,4-DNPH solutions, and color development was observed and recorded.

#### 2.5.2. Confirmation of OM by Nuclear Magnetic Resonance (NMR)

Oxidation of mannan by sodium periodate was also confirmed by 1H-NMR. The 1H NMR was recorded at 500 MHz using Bruker NEO 500 (Bruker, Billerica, MA, USA) equipment. All the peaks were recorded as ppm. The solvent (D_2_O) residual peak at 4.79 ppm was considered as a reference. The OM was freeze-dried to remove the solvent. The freeze-dried powder was then dissolved in deuterium oxide (D_2_O) for NMR evaluation.

### 2.6. Confirmation of Conjugation Reaction between OM and METH Hapten [METH–(KG)_5_]

The reaction between the METH hapten [methamphetamine–(KG)_5_] and OM was carried out at room temperature in the dark overnight. The reaction between OM and the METH hapten [METH–(KG)_5_] was confirmed by LC-MS (Shimadzu LC-MS-2010 EV, Shimadzu, Kyoto, Japan). The absence of the METH hapten from the reaction mixture was considered an indication of a reaction between the two. The hapten peak intensity was observed in LC-MS before mixing with OM and was expected to be reduced or absent upon reaction with OM. A UV spectrophotometer (Shimadzu UV-1800, Shimadzu, Kyoto, Japan) and NMR equipment (Bruker NEO 500, Bruker, Billerica, MA, USA) also confirmed the conjugation between NH2 groups in hapten and aldehyde groups in OM.

### 2.7. Short-Term Stability Assessment of Vaccine Components

The METH conjugated vaccine has three major elements: the METH hapten, OM and the final complete vaccine (METH–(KG)_5_–OM]. A short-term stability study was conducted by storing these vaccine components at −20, 2–8 °C and RT for three months. The stability of METH hapten was evaluated by using LC-MS techniques, and the OM and final vaccine by a UV spectrophotometer. Further, the impact of freeze-drying on the final vaccine formulation was assessed by a UV spectrophotometer.

### 2.8. Immunization of Mice

The ability of the vaccine to induce anti-METH antibodies by was evaluated in mice. The study was approved by the Victoria University Animal Ethics Committee (AEC; protocol number 19/009). The guidelines recommended by the National Health and Medical Research Council Australian Code of Practice for the Care of Animals for Scientific Purposes were followed throughout the animal study. Eight-week-old female C57BL/6 mice were purchased from the Animal Resource Centre (Perth, Australia) for the study. Mice were acclimatized for 5 days and had free access to food and water. The mice were kept in a well-ventilated room at 22 °C with a 12 h light/dark cycle. Animals were then treated with 200 µL METH conjugated vaccine via intraperitoneal injection (IP)). Detail on the animal study timelines is presented in Section 3.6.

#### Enzyme-Linked Immunosorbent Assay

Next, 96-well plates were coated with 45µL METH-BSA (5 ng) antigen and incubated at 37 °C for 18 h. Plates were washed four times with PBS at the end of incubation. A 5% skim milk solution was added and incubated for 30 min to block the plates. The 96-well plates were washed four times with PBS before adding the serum samples. The serum samples were diluted in triplicate on the plate with a starting dilution of 1:40. A set of standard anti-METH monoclonal antibodies (METH monoclonal antibody, MBS531738, My BioSource, San Diego, CA, USA) were also plated in triplicate to prepare a standard calibration curve. Plates were incubated for 1 h at RT. After incubation, the plates were washed 4 times with PBS and 1:10,000 solution of horseradish peroxidase secondary antibody (Goat anti-mouse IgG (H/L): HRP from BIO-RAD-5178–2504) was added to each well of the plates. The plates were incubated for 30 min at RT. Following incubation, plates were washed four times with PBS. A total of 100µL of TMB substrate solution (Thermo Fisher, VIC Australia) was added to each well and incubated at RT for 30 min. The reaction was stopped by adding 100 µL sulfuric acid (0.16 M), and plates were immediately read at 450 nm using a Bio-Rad iMark plate reader.

### 2.9. Statistical Analysis

The antibody data obtained from the animal immunization study were analyzed by GraphPad Prism (version 9.4.0). The student *t*-test model (unpaired or independent *t*-test) was used to check the level of significance between the control and treatment groups. *p* < 0.05 was considered as significantly different.

## 3. Results

### 3.1. Monitoring/Assessment of Reaction between METH and (KG)_5_

The conjugation reaction between (KG)_5_ and METH was confirmed by the disappearance of the METH peak in the reaction sample by TLC and GC. Initially, the sensitivity of the TLC and GC methods was established by running various concentrations of METH standard solution. The TLC method was sensitive (Figure 2A) down to 1 µg/mL METH, which was considered suitable for this experiment. The TLC demonstrated that the METH spot (Figure 2(B-a)) vanished after 24 h of the reaction period (Figure 2(B-b)), suggesting that METH had completely reacted with (KG)_5_. In both solutions, the theoretical concentration of METH was 5 µg when spotted on the TLC plate, which was well above the sensitivity limit of the TLC method. Similarly, in the GC method, the sensitivity of the method was initially established by running various concentrations of METH standard. The method demonstrated sensitivity down to 1.56 µg/mL and was considered suitable for this purpose (Figure 3A,B). METH gave a confirmatory peak at 6.9 min in the GC test before reaction (Figure 3C), and the METH peak (Figure 3D) at 6.9 min was absent in the reaction sample, confirming the completion of the reaction. (KG)_5_ was in excess compared to METH, and upon successful reaction, METH was depleted, and both the TLC and GC test confirmed the depletion of METH in the conjugation reaction.

### 3.2. Confirmation of METH Hapten Identity by LC-MS

The identity of METH hapten was verified by LC-MS. Initially, pure Boc deprotected (KG)_5_ was analyzed by LC-MS in both SIM and scan modes to confirm the suitability of the method. The MW of the (KG)_5_ peptide without Boc is 943.19 amu, and a characteristic peak was obtained in both modes (Figure 4 A,B). Both these tests support the use of the LC-MS techniques to confirm the identity of the METH hapten. The (KG)_5_ is coupled with METH via the reaction between the OH group (c-terminal) of (KG)_5_ and the hydrogen of the secondary amine in METH (Figure 4C).

The molecular weight (MW) of (KG)_5_ without Boc is 943.19 amu, and for METH it is 149.32 amu. In the conjugation reaction, METH hapten with MW of 1074.42 is produced by losing 18 amu as water. To confirm the identity of the METH hapten, the sample was run through LC-MS in SIM and scan modes. In SIM mode, the specification was set to 1074.42(+) and 1073.42(−) (Figure 5A), and for scan mode it was 800–1100 amu (Figure 5B). In both conditions, a confirmatory spectrum at 1075 amu (Figure 5A) was demonstrated by METH hapten. The MS spectra in scan mode are shown in Figure 5B. Further, in this study, we conducted some additional experiments to confirm that this confirmatory peak is not from any other conjugation chemicals. Although we verified the method using standard (KG)_5_, it was also recommended to verify the suitability and specificity of the method using the actual sample matrix. We ran a blank sample (no METH but everything else, Figure 5C), 300 µg METH (Figure 5D) and 1 mg METH sample (Figure 5E). The blank conjugation sample did not demonstrate a peak at 7.4 min, and the rest of the two samples (Figure 5D,E) showed a concentration-dependent peak. This confirmed that the confirmatory peak is from the conjugation between METH and (KG)_5_, and there is no unwanted interference from the conjugation reagents.

### 3.3. Characterization of Oxidized Mannan

Mannan is a biodegradable polysaccharide extracted from yeast. Mannan has two hydroxyl groups at positions 2 and 3 of the ring in a cis position. Mannan in its native form does not have any reactive functional groups to react with amino groups in the METH hapten. Mannan therefore needs to be oxidized by using an oxidizing agent to introduce reactive aldehyde groups on its surface [32,33,34,35]. In the oxidation process induced by sodium periodate, the aldehydes are formed by breaking the bonds between OH groups containing carbon [36,37,38]. A cyclic complex between sodium periodate and mannan leads to the breakdown of the bonds and the production of two-aldehyde moiety for further reaction with amino group [39,40]. In earlier studies, the presence of aldehydes in OM was confirmed by various color and spectrophotometric tests such as UV, FTIR, Schiff reagents and Fehling’s solution [23]. In this study, we explored additional techniques to confirm the presence of aldehydes in OM. 2,4-DNPH is known to react with aldehydes and develop an orange or yellow color or orange-red precipitates. In this study, mannan and oxidized mannan were plated in a TLC silica plate and soaked into a 2,4-DNPH solution. The plates were dried with a heat gun, and color development was observed. The OM developed a strong orange color, while mannan did not develop any color, confirming the presence of aldehydes in OM (Figure 6A,B). The oxidation of terminal mannose residue results in formic acid, which is an aldehyde and carboxylic acid molecule at the same time. The formic acid in OM gives a signature peak at 8.4 ppm in the NMR study. The OM mannan in this study was run through ^1^H NMR, and a characteristic peak at 8.4 ppm (Figure 6C) was observed, confirming that oxidation of mannan was successful.

### 3.4. Confirmation of Hapten–OM Conjugate (Conjugated Vaccine)

The (KG)_5_ linker has five lysines with 10 amino (NH_2_) groups to react with the aldehydes in OM. The aldehydes in OM bonded with amino groups in (KG)_5_ linker through Schiff’s base reaction. As a result, a METH conjugated vaccine [METH–(KG)_5_–OM] was obtained. The conjugation of METH hapten and OM was characterized using LC-MS, a UV spectrophotometer and NMR. In LC-MS, the conjugated sample was run through in SIM mode to confirm the reaction. As demonstrated in Figure 5A, there is a characteristic METH hapten peak at 7.5 min, which disappeared upon reaction with OM (Figure 7A), confirming the conjugation between METH hapten and OM. The conjugation of OM with METH hapten was reconfirmed by UV spectrophotometer analysis. Peptides and proteins exhibit a characteristic UV spectrum at 280 nm due to aromatic ring or disulfide bonds [32,33]. OM was mixed to METH hapten solution at various concentration levels (Figure 7B), and UV absorbance was recorded. The UV spectrum increased with the increase in OM concentrations in the conjugation samples compared to control samples (OM plus PBS). This increase in absorbance at 280 nm indicates the conjugation between amino groups in METH hapten and aldehydes in OM. Further, the reaction between the METH hapten and OM was confirmed by the NMR study. Upon conjugation, the characteristic NMR peak for OM at 8.4 ppm disappeared and confirmed the reaction between OM and hapten (Figure 7C).

### 3.5. Short-Term Stability of the Three Critical Vaccine Components

The METH conjugated vaccine consists of three critical components, namely hapten (METH + KG_5_), OM and the final vaccine (hapten + OM). These three elements are critical concerning the formulation and assessment of their stability. OM, the METH hapten and final vaccine formulations were stored at RT, 2–8 and −20 °C. The stability of hapten was assessed by LC-MS, and the stability of OM and final vaccine formulation was assessed using a UV spectrophotometer. There were no significant changes in LC-MS and UV spectra. All three major elements demonstrated satisfactory stability over the 3-month study period. Further, the effect of freeze-drying on the final vaccine conjugate was assessed, and no impact of freeze-drying was noted regarding the stability of vaccine (Figure 8).

### 3.6. Antibody Response of METH Conjugate Vaccine

After a comprehensive characterization study in the conjugation phase through hapten design, the immunogenic potential of the developed novel vaccine was assessed in a mouse model by administering two doses via intraperitoneal injection (Figure 9A). The collected serum samples were analyzed by ELISA, and results confirmed that the conjugated vaccine is immunogenic. OM was included in this study as a control, and the antibody concentration generated following the second injection was significantly higher in the vaccine group compared to the carrier group, suggesting immunogenicity of the vaccine (Figure 9B). The antibody titer was statistically significant when compared with the control group (*p* < 0.01).

## 4. Discussion

Substances use disorders (SUDs) are an ever-growing challenge globally, and their associated complications are on the rise. Treatment and management of neuropsychological disorders caused by SUDs are always difficult. Immunotherapeutics (vaccines and mAbs) have been considered as potential treatment platforms for SUDs since the early 1970s. Attempts to generate an anti-METH vaccine have been ongoing for 50 years, and significant progress has been made in this field. However, the challenge remains, with no anti-METH vaccine being evaluated in human trials yet. The bottleneck has been the inadequate antibody response produced by vaccine candidates. In the reported studies, METH haptens were designed mainly using either METH or amphetamine and then conjugated with protein carriers such as DT, KLH, BSA and TT with the help of non-immunogenic peptide or alkylating linkers [3,8,14]. Preclinical studies using anti-METH vaccines demonstrated a varied amount of antibody response and were not effective in protecting the animals from the effects of METH, leaving a successful immunotherapeutic lacking [14]. One of the main success criteria of these vaccines is to be able to produce antibody responses with high affinity or concentration in order to stop the entry of METH into the CNS and accordingly prevent its action and adverse events. This was supported by several clinical trial outcomes involving morphine and nicotine vaccines; however, these trials were unsuccessful due to insufficient antibody production [3,8,16]. Based on this previous experience, new avenues are being explored to stimulate strong antibody responses to provide complete protection against METH effects. Amongst all the strategies, the use of adjuvants, strong immunogenic carriers, nanocarriers and hapten design with superior properties is a key approach to increase antibody production.

As the METH molecule itself is not immunogenic, it requires an immunogenic carrier to activate an antibody response. Development of anti-METH vaccines, where the surface of the METH or amphetamine is modified to attach the linker first and then add an immunogenic carrier, is highly chemistry-driven. To introduce the linker, the modifications are made at various positions of the methamphetamine or amphetamine such as the phenyl position, ortho-meta positions of an aromatic ring or the primary or secondary amine or N-methyl. During the hapten design, precautions are taken to reserve the actual structure of the drug molecule as much as possible to ensure better specificity. Hence, although amphetamine is more frequently used to synthesize hapten in this study, METH HCl was chosen to develop the hapten using the peptide linker and (KG)_5_ and OM mannan as an immunogenic carrier.

The conjugation reaction between OM and peptide is a highly chemistry-driven approach and is considered more complicated than its conjugation to protein carriers due to the interaction with lysine residues. Proteins are large structures with a large amount of lysine, making reactions very efficient and complete. However, short peptide linkers often do not contain significant lysine, and conjugation efficiency is limited. A peptide linker with adequate lysine residue is required for an efficient conjugation reaction. Based on previous experience, a short peptide linker with five-lysine and five-glycine amino acids (KG)_5_ has been used to ensure efficient reaction between aldehydes in OM and amino groups in the peptide linker [41]. In this study, (KG)_5_ was used during the METH hapten synthesis through hapten design. As this is a novel platform, the assessment of conjugation efficiency by using suitable, sensitive and robust analytical techniques is crucial. In this study, various analytical techniques such as TLC, GC and LC-MS were used to confirm the reaction between METH and (KG)_5_, and the identity of the hapten was confirmed by LC-MS. All these methods were initially verified for their suitability for the current application and then used for actual sample analysis. All the analytical techniques confirmed the formation of hapten, which was then conjugated with OM. One of the key features of this study is the use of OM as an immunogenic carrier, which plays a critical role in this novel platform. OM was selected for use in this METH–(KG)_5_–OM conjugate for several reasons (i) the OM as carrier will deliver the METH hapten to APCs (dendritic cells and macrophages), (ii) the matrix may contain non-reacted aldehyde groups necessary to allow the escape of peptides from endosomes within dendritic cells [42,43] and (iii) the intact non oxidized mannan residues can induce immune response via TLR4 and mannose receptor activation [44]. The use of OM has already demonstrated significant efficacy in producing strong antibody responses in various vaccine formulations [45,46]. In this study, we oxidized mannan to introduce reactive aldehyde groups on its surface. The presence of aldehydes was confirmed by 2,4-DNPH and NMR tests. UV, NMR and LC-MS techniques were also used to evaluate its conjugation with METH hapten. All these techniques confirmed the formation of OM and its conjugation with METH hapten. The immunogenic potential of this novel anti-METH vaccine was evaluated in C57BL6 mice, and the candidate demonstrated immunogenic potential. OM was included in this study as a control, and antibodies generated by the vaccinated group were significantly higher than the OM group (*p* < 0.01). In previous pilot studies, reduced mannan was also used as a carrier; however, OM was shown to induce superior antibody responses (data not shown), and as such OM was used in these studies.

Further, in this study, we evaluated the stability of some critical components of the vaccine formulation to assess commercial viability. The hapten [METH+(KG)_5_], OM and final conjugate [METH+(KG)_5_+OM] were stored at room temperature, 2–8 and −20 °C, and their stability was assessed by LC-MS and UV at predefined time intervals. All the critical vaccine components demonstrated stability at all the storage conditions, suggesting commercial viability. In many resource-limited countries, vaccine storage and transportation are critical issues, and the stability of the vaccine at room temperature will be of utmost importance. However, long-term stability studies will be required to assess the actual shelf life and storage conditions beyond 3 months. Further, the impact of freeze-drying on the final vaccine was evaluated. Freeze-drying is often beneficial for pharmaceutical products, especially heat-sensitive products in terms of stability, storage and transportation [47,48]. The freeze-dried powder has superior stability compared to liquid form and can be stored for an extended period. However, freeze-drying may also cause instability of the final product due to structural change [49]. In our study, freeze-drying did not have any negative impact on the stability of the final vaccine and provided an opportunity to store the vaccine in powder form.

The use of peptide linkers for the assessment of immunogenic potential has been reported previously [14,19,50] as a reasonable and effective means of preventing METH biodistribution and METH-induced locomotor activity. However, previously reported studies were conducted using protein carriers, and in the current study we used a carbohydrate (OM) as an immunogenic carrier. Hence, it is difficult to make a direct comparison between a well-established protein-carrier-based conjugated vaccine system and this novel carbohydrate-based approach.

Previously reported studies also used short-length peptide linkers and demonstrated that monoamine or diamine linker provides efficient binding and orientation of the hapten molecule and is thus important for an immunogenic response [10]. In our study, a medium-length [KG)_5_] peptide linker with 10 amino acids was used. Its effectiveness has been widely investigated in cancer and multiple sclerosis vaccine models [29]. It has also been reported that with OM as carrier, no adjuvants are required to produce strong immune responses. Hence, this novel platform has several new aspects compared to the existing protein-based anti-METH vaccine platforms. One can further optimize antibody responses by using self-adjuvanting tricomponent vaccines that include antigen–glycoprotein–universal T helper epitope PADRE and Pam_3_CysSer adjuvant [51,52,53] or the use of nanoparticles [54,55,56]. Previous research on the efficacy and stability of relevant candidate MS vaccine using oxidized mannan conjugated with antigenic epitope peptide MOG35–55 is in line with and supports the findings of this study [57,58].

However, the present study exhibits an encouraging outcome for a novel platform. The important feature of this study was that the various critical intermediate steps of the vaccine development were confirmed by several analytical tools and will potentially be useful in future research. The novel anti-METH vaccine has been shown to be immunogenic in mice. The extensive characterization studies and the animal study result supports its suitability as a candidate for further exploration to aid in the treatment of METH addiction. However, being a novel platform, this study has some limitations, which need to be addressed in future study design to generate more meaningful information and make significant progress. This includes the effect of adjuvants, length of peptide linkers, hapten copy attached to the OM, route of administration and candidates’ ability to stop METH entry into the central nervous system. In addition, large-scale studies to generate substantial data will be essential. Long-term stability study of the vaccine component and vaccine formulation should also be investigated to generate real-time information. Relapse is a critical issue in the treatment and management of SUDs. The relapse rate is quite high after residential rehabilitation. Hence, a successful vaccine can play critical role in preventing the relapse by providing long-term protection in the event of taking the drug accidentally or intentionally. The effectiveness of the proposed vaccine should also be investigated by assessing their effectiveness to prevent relapse.

## 5. Conclusions

A novel anti-METH conjugated vaccine was developed by using METH itself as a drug candidate, (KG)_5_ as a peptide linker and OM as an immunogenic carrier. The various phases of the METH conjugated vaccine development were characterized by a set of comprehensive analytical techniques. The animal study also confirmed that the developed novel vaccine is immunogenic. Further, a short-term stability study demonstrated satisfactory stability of the major three vaccine components. The important aspect of this study is the use of OM as an immunogenic carrier for the anti-drug vaccine. OM has been used for many vaccine formulations previously, and now in this study it has proven to be immunogenic. Both (KG)_5_ linker and OM are unique to the anti-METH vaccine development strategy and demonstrated excellent promise in this study. The present study outcome is encouraging and will form the basis of future improvement studies as well as preclinical and clinical studies.

## Figures and Tables

**Figure 1 vaccines-11-00340-f001:**
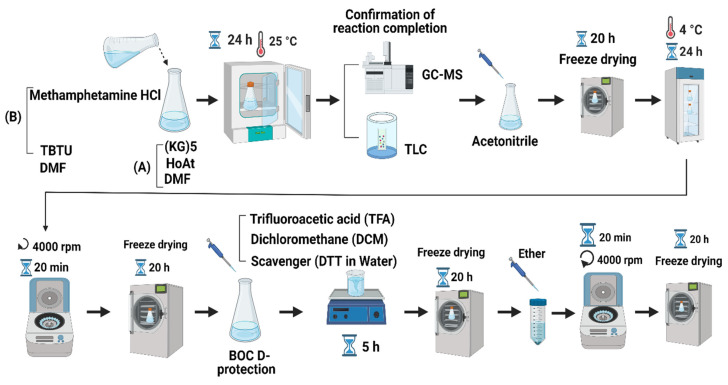
Summary of METH hapten synthesis protocol (figure created with BioRender.com).

**Figure 2 vaccines-11-00340-f002:**
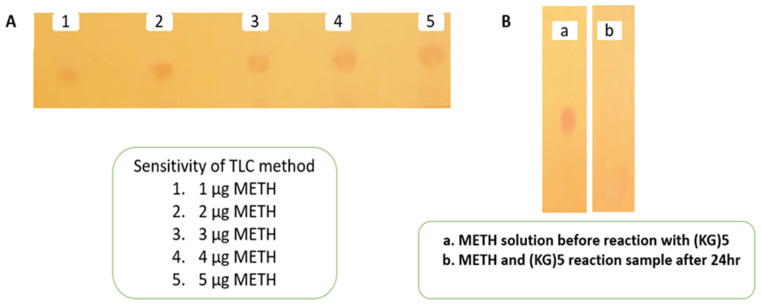
Monitoring of reaction completion between METH and (KG)_5_ by TLC. (**A**) Sensitivity of TLC method, (**B**) confirmation of reaction between METH and (KG)_5_.

**Figure 3 vaccines-11-00340-f003:**
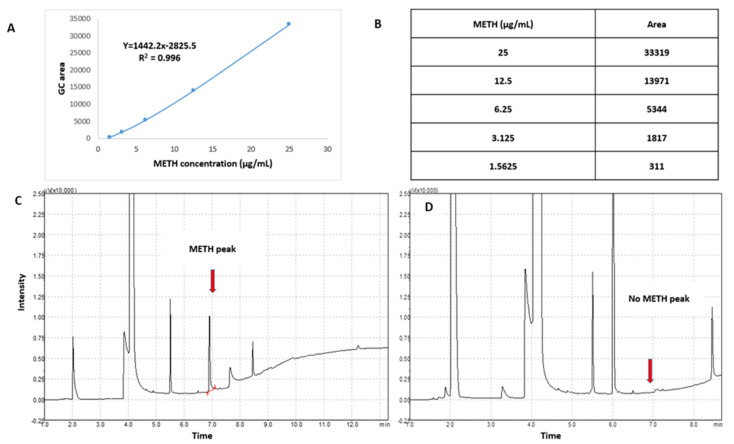
Monitoring of reaction between METH and (KG)_5_ by GC. (**A**) METH standard calibration curve, (**B**) METH concentration levels for the calibration curve, (**C**) METH peak before reaction with (KG)_5_, and (**D**) disappearance of METH peak after reaction with (KG)_5_.

**Figure 4 vaccines-11-00340-f004:**
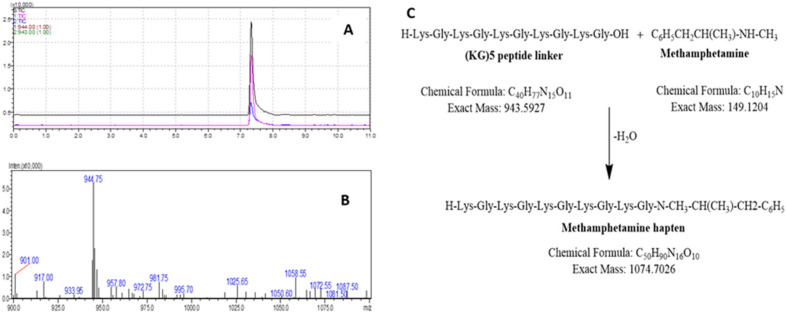
LC-MS method suitability check for METH hapten analysis. (**A**) Representative chromatogram of (KG)_5_ standard in SIM mode, (**B**) Representative mass spectrum of (KG)_5_, (**C**) Reaction scheme between METH and (KG)_5_.

**Figure 5 vaccines-11-00340-f005:**
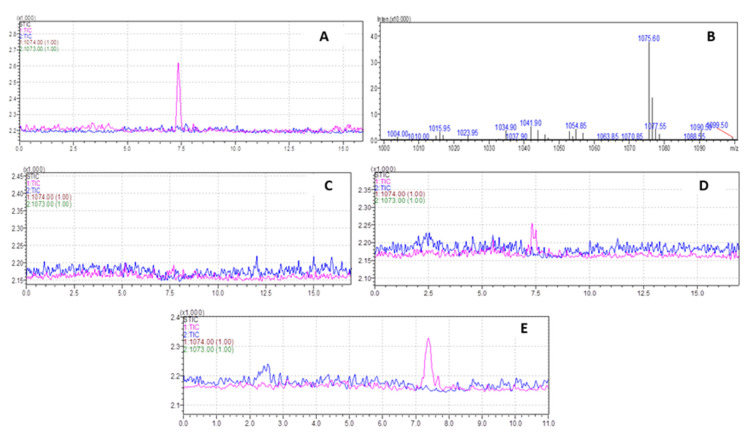
Characterization of METH–(KG)_5_ conjugates by LC-MS. (**A**) METH–(KG)_5_ in SIM mode, (**B**) METH–(KG)_5_ in scan mode, (**C**) blank reaction sample MS spectrum, (**D**) MS spectrum with 300 µg METH in reaction and (**E**) MS spectrum with 1 mg METH in reaction.

**Figure 6 vaccines-11-00340-f006:**
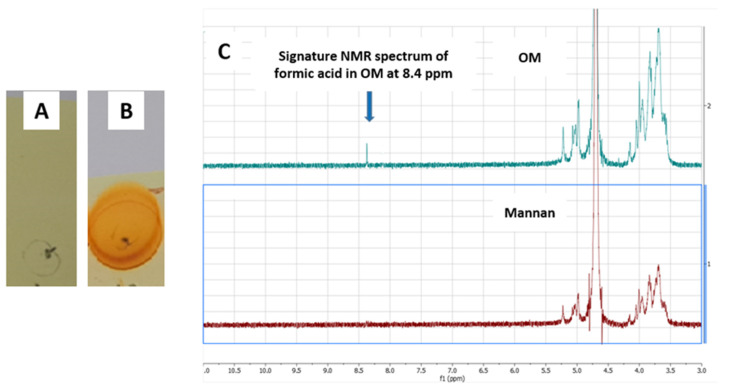
Confirmation of OM by 2,4-DNPH test and NMR study. (**A**) Mannan with 2,4-DNPH solution, (**B**) OM with 2,4-DNPH solution and (**C**) NMR spectrum of mannan and OM.

**Figure 7 vaccines-11-00340-f007:**
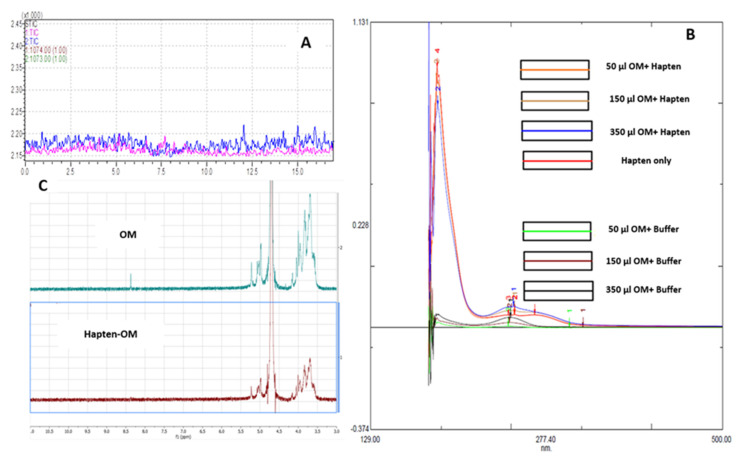
Confirmation of final vaccine by LC-MS, UV spectrophotometer and NMR. (**A**) Disappearance of METH hapten peak, (**B**) reaction titration between METH hapten and OM, (**C**) NMR spectrum of OM and hapten––OM conjugate.

**Figure 8 vaccines-11-00340-f008:**
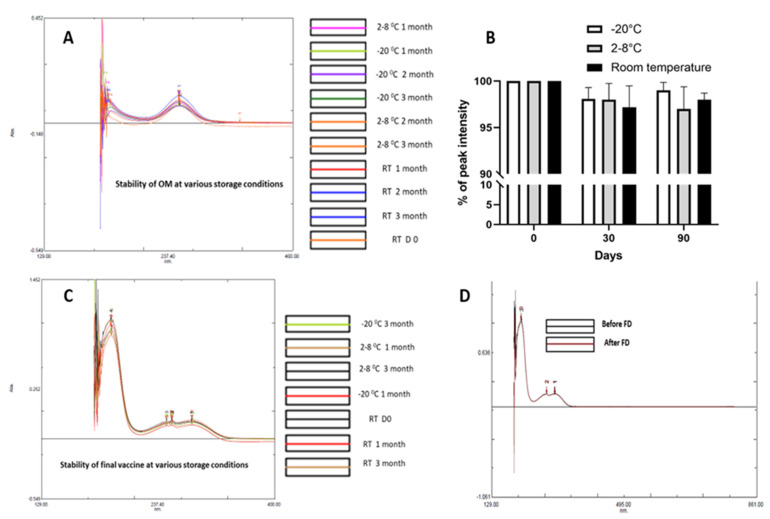
Assessment of the stability of various vaccine elements at various storage conditions. (**A**) OM, (**B**) hapten, (**C**) final vaccine and (**D**) effect of freeze-drying.

**Figure 9 vaccines-11-00340-f009:**
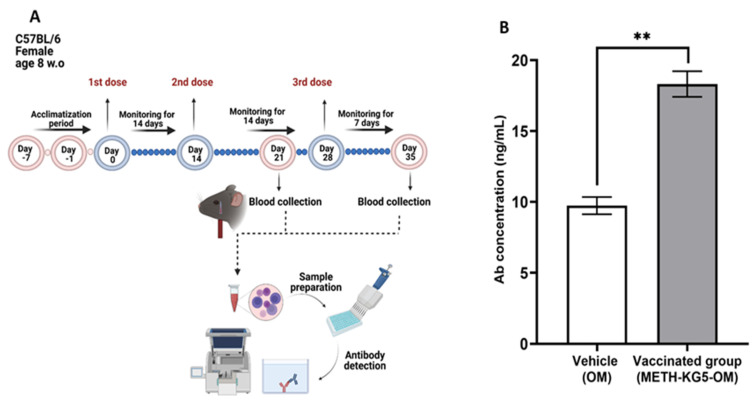
(**A**) Animal study timeline and (**B**) antibody titer by ELISA. ** *p* < 0.01; bars represent mean + standard error of the mean (SEM).

## Data Availability

Not applicable.

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
