# Peer review of "Development of Methamphetamine Conjugated Vaccine through Hapten Design: In Vitro and In Vivo Characterization"

_vaccines, 2023, doi:10.3390/vaccines11020340_

Round 1
Reviewer 1 Report
The manuscript titled “Development of methamphetamine conjugated vaccine through hapten design: in vitro and in vivo characterization” addressed the relevant threat from rising Substances abuse disorders (SUDs) by designing a conjugated vaccine comprising methamphetamine (METH) HCl as hapten, (Lysine-Glysine) as a peptide linker, and oxidized-mannan (OM) as an immunogenic carrier. The study portrayed significant scientific rigor and novelty in terms of using OM as an immunogenic carrier in a glyco-conjugated vaccine against SUD. The methodology and validations are comprehensive and well demonstrated.
The study will certainly encourage sustainable vaccine-design strategies against SUDs and therefore can be recommended for publication with minor clarifications.
There are minor comments which can be addressed:
1. A spelling and grammar check is recommended throughout the manuscript to correct minor errors e.g., Line 23- linker is misspelled
2. How many vaccine constructs were designed to finalize the most suitable construct? Was there any pre-defined statistical criteria/ model or a prior reference study? Please address and include in the manuscript
3. The authors can include a reference vaccine candidate (previously reported) to quantitatively discuss the relative advantage of using the proposed conjugated vaccine (in terms of cost and efficacy)
4. One of the major concerns to vaccines against SUDs is the relapse. What is the authors’ recommendation on enhancing the period of vaccine efficacy? This can be included in discussion
Author Response
The manuscript titled “Development of methamphetamine conjugated vaccine through hapten design: in vitro and in vivo characterization” addressed the relevant threat from rising Substances abuse disorders (SUDs) by designing a conjugated vaccine comprising methamphetamine (METH) HCl as hapten, (Lysine-Glysine) as a peptide linker, and oxidized-mannan (OM) as an immunogenic carrier. The study portrayed significant scientific rigor and novelty in terms of using OM as an immunogenic carrier in a glyco-conjugated vaccine against SUD. The methodology and validations are comprehensive and well demonstrated.
The study will certainly encourage sustainable vaccine-design strategies against SUDs and therefore can be recommended for publication with minor clarifications.
Authors Response: Thank you so much for your time to review the manuscript and suggestions to improve the quality of the manuscript. The authors have addressed your recommendations to improve the manuscript's quality.
- A spelling and grammar check is recommended throughout the manuscript to correct minor errors e.g., Line 23- linker is misspelled
Authors Response: The article has been checked by two native English speakers to improve the grammar and spelling.
- How many vaccine constructs were designed to finalize the most suitable construct? Was there any pre-defined statistical criteria/ model or a prior reference study? Please address and include in the manuscript
Authors Response: Thank you so much for the comment. No statistical model was used. The authors are doing a very early-stage investigation of a novel platform. The initial objective was to conduct a proof of concept study using OM as an immunogenic carrier to confirm that the conjugated vaccine can be developed and it’s immunogenic. In this project initially, we developed a vaccine using Amphetamine as starting material, and in this study, we have sued methamphetamine as starting material to develop the vaccine. Our intention was to get a lead and then adopt the DoE approach to harness this platform's critical variables to optimize the formulation. Recently we have also finished two more projects (unpublished) including nano-vaccine formulation and vaccine with short peptide linkers. Altogether we have developed six candidates as of now and hopefully, we will get a “lead” from here to further optimize the vaccine using statistical/DoE strategy.
- The authors can include a reference vaccine candidate (previously reported) to quantitatively discuss the relative advantage of using the proposed conjugated vaccine (in terms of cost and efficacy)
Authors Response: In consistent with our response in point 2, we are doing a very early stage investigation of a novel platform and we are not in a position to make a direct comparison with previously reported vaccines. The previously reported vaccine has used proteins as an immunogenic carrier and are optimizing that protein-based platform for the last 3-4 decades. However, as stated earlier, we are getting some lead to design further experiments to explore all the critical variables and we may get some better candidates to make a direct comparison with previously reported protein carrier-based vaccines.
- One of the major concerns to vaccines against SUDs is the relapse. What is the authors’ recommendation on enhancing the period of vaccine efficacy? This can be included in discussion
Authors Response: Thank you so much for the excellent comment. The authors have added some statements in the discussion section.
Reviewer 2 Report
Dear Authors
It was with great pleasure that I reviewed your manuscript.
It is very well structured.
The introduction is well-reasoned.
The methods and materials used have been well described.
The results are well presented.
The discussion substantiates the results well.
My Best Regards
Author Response
Dear Authors
It was with great pleasure that I reviewed your manuscript. It is very well structured. The introduction is well-reasoned. The methods and materials used have been well described. The results are well presented. The discussion substantiates the results well.
My Best Regards
Authors Response: Dear Reviewer, Thank you so much for the time to review the manuscript.
Reviewer 3 Report
Overall this is a well written paper with an interesting result on the vaccines area
INTRODUCTION
The introduction provides sufficient background information for readers to understand the research aim, however the authors should clarify the importance of METH and the actual knowledge in this area. It seems like some important clarification is missing.
The introduction provides a good perspective of the main topic, however, to make the introduction more substantial, the author may wish to provide several actual references.
Motivations for this study are more than clear but the objectives must be clearly defined at the Introduction.
METHODS
The methodology proposed to reach the aim of the study look appropriate, well designed and conducted.
Specify the manufactured, company and country of all the instrument used in the research.
There are a few instances where assertions are made that are not substantiated with references.
RESULTS
Results paragraphs include the most relevant data.
Figures are correct
DISCUSSION
All possible interpretations of the data considered are consistent, and in line with study aims
Conclusion should respond the research aim
Explain limitation of the study and future research line according to the study conclusion
Include practical application of the results
Author Response
Overall this is a well written paper with an interesting result on the vaccines area
INTRODUCTION
The introduction provides sufficient background information for readers to understand the research aim, however the authors should clarify the importance of METH and the actual knowledge in this area. It seems like some important clarification is missing.
The introduction provides a good perspective of the main topic, however, to make the introduction more substantial, the author may wish to provide several actual references.
Motivations for this study are more than clear but the objectives must be clearly defined at the Introduction.
Authors Response: Thank you so much for your recommendation on the introduction. The authors have added some relevant references and tried clarifying the study’s objective.
METHODS
The methodology proposed to reach the aim of the study look appropriate, well designed and conducted.
Specify the manufactured, company and country of all the instrument used in the research.
There are a few instances where assertions are made that are not substantiated with references.
Authors Response: The authors have included the instrument details as suggested and included references to support the claim.
RESULTS
Results paragraphs include the most relevant data.
Figures are correct
Authors Response: Thank you so much for your comments
DISCUSSION
All possible interpretations of the data considered are consistent, and in line with study aims
Conclusion should respond the research aim
Explain limitation of the study and future research line according to the study conclusion
Authors Response: Thank you so much for your suggestions. The authors have added some more discussion around the limitations of the study and provided more information in the conclusion to respond to the research aim.